# Dynamic Semi IPNs with Duple Dynamic Linkers: Self-Healing, Reprocessing, Welding, and Shape Memory Behaviors

**DOI:** 10.3390/polym13111679

**Published:** 2021-05-21

**Authors:** Yanning Zeng, Weiming Yang, Shuxin Liu, Xiahui Shi, Aoqian Xi, Faai Zhang

**Affiliations:** Key Laboratory of New Processing Technology for Nonferrous Metal and Materials (Ministry of Education), College of Material Science and Engineering, Guilin University of Technology, Guilin 541004, China; ywm18378330339@163.com (W.Y.); sxliu12138@163.com (S.L.); sxh06070521@126.com (X.S.); Xaq000109aqx@163.com (A.X.); zhangfaai@163.com (F.Z.)

**Keywords:** CANs, semi IPNs, self-healing

## Abstract

Thermoset polymers show favorable material properties, while bringing about environmental pollution due to non-reprocessing and unrecyclable. Diels–Alder (DA) chemistry or reversible exchange boronic ester bonds have been employed to fabricate recycled polymers with covalent adaptable networks (CANs). Herein, a novel type of CANs with multiple dynamic linkers (DA chemistry and boronic ester bonds) was firstly constructed based on a linear copolymer of styrene and furfuryl methacrylate and boronic ester crosslinker. Thermoplastic polyurethane is introduced into the CANs to give a semi Interpenetrating Polymer Networks (semi IPNs) to enhance the properties of the CANs. We describe the synthesis and dynamic properties of semi IPNs. Because of the DA reaction and transesterification of boronic ester bonds, the topologies of semi IPNs can be altered, contributing to the reprocessing, self-healing, welding, and shape memory behaviors of the produced polymer. Through a microinjection technique, the cut samples of the semi IPNs can be reshaped and mechanical properties of the recycled samples can be well-restored after being remolded at 190 °C for 5 min.

## 1. Introduction

Polymeric material is conventionally categorized as thermoset or thermoplastic, which is based on whether the material flows at elevated temperatures. Traditionally, thermoset polymer with irreversible covalent linkages shows excellent mechanical properties, creep resistance, chemical/solvent stability, and ideal optical performance, applications in coatings, adhesives, and biomedical materials, however, it cannot be reprocessed or reshaped once its network is formed [1,2]. Thermoplastic polymer with reprocess ability, can flow when heated, which permits extrusion, injection, and molding, whereas normally it exhibits poor chemical and solvent resistances, low mechanical strength, and low dimensional stability at high temperatures. To combine both merits of thermoset and thermoplastic polymers, covalent adaptable networks (CANs) are explored; these can provide a stimulus responsive (such as temperature, light, electric, pH, moisture, and chemicals) to the polymer with fantastic properties such as shape memory, repairing, crack healing and stress relief [3,4,5]. CANs represent a wide range of polymer networks with dynamically exchangeable covalent bonds. When dynamic covalent bonds are activated at high temperatures, the rapid exchange reactions provide a rearrangement of the networks, enabling the polymer to be deformed, processed, and recycled, then the shape can be fixed by quenching the exchange reactions at low temperatures [6,7]. CANs can be identified into two types in terms of their exchange mechanisms. One of the CANs employs a dissociative cross-link exchange mechanism such as DA chemistry; covalent bonds are firstly broken and then formed again at a different position. The other CANs apply associative exchange bonds between polymer chains, in which the original cross-link is only broken when a new covalent bond in another place has been formed, making the networks permanent as well as dynamic. Based on associative exchange mechanisms, polymer networks characterize a constant value of chemical bonds, and cross-links do not depolymerize at high temperatures [8]. To extend the realm of associative CANs, Leibler and co-workers prepared permanent polyester/polyol networks with a gradually decreased viscosity upon heating, similar with vitreous silica, hence, the name of vitrimers for those materials has been proposed [9]. Vitrimers possess a unique combination of properties, including recyclability, creep resistance, enhanced adhesion, adjustable solid-to-liquid transition, environmental stress-cracking resistance, shape memory, and self-healing ability. Currently explored vitrimers are demonstrated, with a subdivision in terms of the nature of dynamic associative exchange reactions such as carboxylate transesterification [10], transamination of vinylogous urethanes [11], transalkylation [12], siloxane silanol exchange [13], olefin metathesis exchange [14], disulfide exchange [15], imine metathesis [16], boronic ester exchange [17], silyl ether exchange [18] etc. Despite the great scientific significances of these systems, their employment in practical industrial applications is far from being straightforward. Because excellent dynamic properties are obtained at the expense of mechanical performance, CANs are present in the form of elastomers and gels. Besides, reprocessing, and remolding ability of the CANs have been demonstrated, almost via hot press (high pressure and temperature), and rigorous conditions are required.

Reinforcement of fracture toughness has been challenging in the field of elastomers and gels [19,20]. Materials fracture mainly results from the localized stress concentration in a molecular network under deformation; simple incorporation of a stress-dissipation device in network structure can suppress the stress concentration, leading to high-toughness materials [21]. Interpenetrating polymer networks (IPNs) hydrogel composed of two kinds of networks with different strengths has been explored for high mechanical properties. Differently, semi IPNs are composed of host networks and guest polymers which do not form permanent cross-links [22,23,24]. Various physical properties (thermal properties and mechanical toughness) of semi IPNs can be modified or improved by introduction of guest polymers which have various thermal or flow properties. Moreover, the compatibility between two component polymers is important to obtain homogeneous material with fantastic properties. Generally, to fabricate IPNs/semi IPNs, two approaches are involved: sequential and simultaneous networks. The former indicates the guest polymers preparation in the presence of the first networks, while the latter means independent synthesis of both guest polymers and host networks at the same time. Remarkably, numerous studies have been reported which relate to the fabrication of IPNs and semi IPN materials [25,26,27,28].

Thermoplastic polyurethane (TPU) elastomer is a linear segmented block copolymer which forms by the reaction of diisocyanates, oligomeric diols, and chain extender (low molecular weight diols). TPU is made up of a polyol soft segment and hydrogen bonded hard segments. Low melting soft segments, rigid hard segments and a flexible connection provide an (AB)n-type segmental copolymer and the content of hard segments can be regulated to gain optimal mechanical properties [29,30,31]. A wide range of physical properties for TPU can be obtained by change molecular structure, molecular weights, and its distributions in each segment, and the ratios of soft/hard segment. Moreover, TPU can be compatible with boronic ester due to a soft segment of polyester or polyether.

To the best of our knowledge, there is no report about semi IPNs based on CAN matrices and TPU as guest polymers. In this work, copolymers P(St-*co*-FMA) of styrene (St) and furfuryl methacrylate (FAM) are firstly prepared to provide a furan functional group that is similar with reported works [32,33]. Then we demonstrate the synthesis of semi IPNs by a sequential reaction approach using P(St-*co*-FMA), boronic ester crosslinker and TPU elastomer, along with the high compatible networks. It is observed that the introduction of TPU into CANs results in a remarkable increase in flexibility and the obtained semi IPNs can be facilely reprocessed by a microinjection technique. Meanwhile, the semi IPNs display superior mechanical toughness and thermal peripteries. Furthermore, the addition of TPU and boronic ester crosslinker are investigated to regulate the peripteries of the produced semi IPNs. This work might contribute a promising pathway to enhance the properties of CANs and extend their application.

## 2. Experimental

### 2.1. Materials

Thermoplastic polyurethane (TPU-1170A) was purchased from BAYER Co., Ltd Leverkusen, Germany. Furfuryl methacrylate (99%) (FMA) was supplied by Tokyo Chemical Industry, Japan. Benzene-1,4-diboronic acid (97%), 4-dimethylaminopyridine (99%), styrene (St), Furan, ethyl acetate, potassium ethylxanthate, 3-amino-1,2-propanediol, maleic anhydride, phenylboronic acid, 1, 2-propanediol, cuprous chloride, ethyl-2- bromoisobutyrate, ethyl α-bromoisobutyrate (EBiB), 1,1,4,7,10,10-hex amethyltriethylene tetramine (HMTETA), cuprous chloride (CuCl) and *N*-(2,3-dihydroxypropyl) maleimide (DHPM) were supplied by Aladdin, Shanghai, China. Tetrahydrofuran (THF), heptane, *N, N*-dimethylformamide, methanol, ethanol, toluene, acetone, and magnesium sulfate (MgSO_4_) were purchased from XiLong scientific Co., Ltd. (Shantou, China).

### 2.2. Synthesis of 2,2′-(1,4-Phenylene)-bis [4-Methyl-1,3,2-Dioxaborolane] (BDB)

As reported [34], BDB was synthesized benzene-1,4- diboronic acid (3.0 g, 18.1 mmol) and 1,2-propanediol (2.82 g, 37.1 mmol) were mixed in THF (30 mL) and water (0.1 mL). MgSO_4_ (5 g) was added. After 24 h at room temperature, the solution was filtered and concentrated under reduced pressure to obtain the target compound as a slightly yellow solid. Then, the solid was immersed into heptane and stirred for 1 h at 50 °C, filtered and concentrated under reduced pressure to obtain the target compound BDB as a white solid (5.39 g, yield: 92.6%).

### 2.3. Synthesis of 1-[(2-Phenyl-1,3,2-Dioxaborolan-4-Yl)Methyl]-1H-Pyrrole-2,5-Dione (DHPM-B)

A toluene dissolution (70 mL) of DHPM (5.5 g, 23.1 mmol) and phenylboronic (2.8 g, 23.1 mmol) was transferred into a three-necked flask with a reflux at 135 °C for 6 h. Under reduced pressure, the solvent was removed and finally a light white powder (6.75 g; yield: 81.3%) was obtained with extraction from its ethanol solution as the reported work [35]. 

### 2.4. Copolymerization of FMA and St (P(St-co-FMA)) 

General procedure: A 50-mL round-glass flask was charged with a toluene solution of FMA (10 g, 60 mmol), St (4.17 g, 40 mmol), HMTETA chain transfer agent (0.23 g, 1 mmol), CuCl catalyst (0.198 g, 2 mmol) and EBiB ligand (1.95 g, 10 mmol) with the ratio of (60:40:1:2:10). The polymerization was carried out under N_2_ atmosphere at 90 °C for 18 h, then it was quenched with methanol. The P(St-*co*-FMA) copolymer was washed with methanol, then dried under reduced pressure at 90 °C for 18 h, and weighed (9.88 g; yield: 69.7%).

### 2.5. Synthesis of CANs (CPSF CANs)

General procedure: A THF dissolution of P(St-*co*-FMA) (5 g) and DHPM-B (0.3 g, 0.9 mmol) was transferred into a three-necked flask, stirred at 60 °C for 30 min, then added crosslinker 6% BDB (0.153 g, 0.9 mmol), and kept in an oven at 120 °C for 16 h. The CANs 6%CPSF was obtained as in Figure 1, and x% is the amount of added crosslinker agent BDB.

### 2.6. Fabrication of Semi-IPN with CPSF and TPU (semi IPNs of CPSFTPU)

General procedure: A DMF solution of P(St-*co*-FMA) (5 g), TPU (5 g), DHPM-B (0.6 g, 1.8 mmol) and crosslinker BDB (0.3 g, 1.8 mmol) was heated to remove the solvent at 60 °C for 24 h, then kept in an oven at 120 °C for 16 h for further reaction. Then the semi IPNs 6%CPSFTPU_1_ was obtained as Figure 2, x% is the addition amount of crosslinker BDB and the value of subscript is ratio of P(St-*co*-FMA)/TPU.

### 2.7. Self-Healing, Shape-Memory, and Reprocessing

Self-healing was conducted by cutting the sample (semi IPNs 6%CPSFTPU_1_) into two parts and putting them together for healing at varied temperatures for different times. Healing efficiencies were determined by the ratio of tensile strength/breaking strain of the healed samples and those of the original sample.

To study the shape memory capability, the cured semi IPNs 6%CPSFTPU_1_ was checked. The strip sample was put into an oven at 120 °C and was bent into various shapes by an external force; finally, it was cooled down to room temperature. We recorded the digital photos of the strip sample before and after reshaping.

The reprocessing sample (semi IPNs 6%CPSFTPU_1_) was cut into small pieces. Mini twin screw injection molding machine (WLG10G, Shanghai Xinshuo Precision Machinery Co., Ltd., Shanghai, China) as a reprocessing equipment, obtained sample particles were put into the barrel, then post-processing was performed at 190 °C, and the samples were repeated for three cycles.

### 2.8. Characterizations

Fourier transform infrared (FTIR) spectra were collected by the KBr tablet methods using a Nicolet 205 FTIR spectrometer (Madison, USA) from 600 to 4000 cm^−1^. Nuclear magnetic resonance (^1^H NMR) spectra of BDB, DHPM-B and P(St-*co*-FMA) were recorded at room temperature on an AV500 spectrometer (Bruker, Germany). Deuterated Chloroform (CDCl_3_) and dimethyl sulfoxide-*d_6_* was used as the solvent.^1^H NMR chemical shifts were referenced to the CDCl_3_ signals. Thermogravimetry analysis (TGA) with a heating rate of 10 K/min in nitrogen atmosphere from 35 to 800 °C on a TA Q500 (Milford, USA) was employed to examine the thermal decomposition behavior of the P(St-*co*-FMA), CPSF and CPSFTPU series. Dynamic mechanical analysis (DMA) and stress relaxation tests were carried out using a TA Q800 instrument (Milford, USA). Stress relaxation experiments were conducted by monitoring the stress decay at a constant strain of 1% after equilibrating at the required temperatures for 20 min. The mechanical performance test employed the UTM4503SLXY universal tensile testing machine of Shenzhen Sansi aspect Technology Co., Ltd., ShenZhen, China, with 2 mm/min tensile rate was utilized to perform the mechanical test. Young’s modulus, breaking strain and tensile stress were obtained by mechanical performance test. Scanning electron microscopy (SEM) was conducted using a field emission scanning electron microanalyzer (HITACHI, S-4800, Tokyo, Japan) at an acceleration voltage of 10 kV. An optical microscopy (BA210, Motic China Group Co., Ltd., Hong Kong, China) was used to evaluate the self-healing behaviors. The density of polymer was measured by TWS series touch screen electronic densitometer (TWS-300S, Mzkeyi Co., Ltd., ShenZhen, China).

As shown by Equations (1) and (2), the sol fraction and swelling ratio were determined by equilibrium swelling experiment based on the Flory–Rehner equation. It was performed by immersing polymer in toluene, THF, heptane or acetone at room temperature for 72 h, the fresh solvent replaced the old one each 24 h, then solvent was quickly removed by a filter paper. The samples were weighed immediately, and dried in a vacuum oven at 60 °C until constant weight. Three time-points were used to check each sample. Assuming that the initial mass was $m_{0}$, the mass after swelling was $m_{1}$, and the mass after drying was $m_{2}$, it can be obtained:(1)$$\mathrm{Swelling}\;\mathrm{ratio}\;\mathrm{is}\;\mathrm{defined}\;\mathrm{as}:\;\frac{m_{1}-m_{2}}{m_{2}}$$
(2)$$\mathrm{Sol}\;\mathrm{fraction}\;\mathrm{is}\;\mathrm{determined}\;\mathrm{as}:\;\frac{m_{0}-m_{2}}{m_{0}}$$

The crosslinking density (*C*_d_) of produced polymer was determined by the equilibrium swelling method with toluene as solvent. The polymer was cut into five samples, 20 mm × 10 mm × 0.6 mm, weighed, and immersed in a separate bottle containing 50 mL toluene for five days. After equilibrium swelling was achieved, the sample was dried between the sheets and weighed again. The *C*_d_ of the CSPF was calculated by Flory–Rehner Equation (3) [36,37].
(3)$$C_{d}=-\frac{ln\left[\left(1-V_{r}\right)+V_{r}+\chi V_{r}^{2}\right]}{V_{s}\left(V_{r}^{1/3}-V_{r}/2\right)}$$
(4)$$V_{r}=\frac{m_{0}/\rho_{0}}{m_{0}/\rho_{0}+\left(m_{1}-m_{0}\right)/\rho_{c}}$$
(5)$$\;\;\;\chi =0.487+0.228V_{r}$$
where *m_0_* is the initial mass of the sample; *m*_1_ is the mass of the sample after swelling equilibrium; *ρ*_c_ is the density of toluene (25 ℃); *ρ*_0_ is the density of polymer; *V*_r_ is the volume fraction of polymer, the calculation formula is Equation (4), *χ* is the interaction parameter between solvent and polymer, the simplified calculation formula is Equation (5); *V*_s_. is the molar volume of the solvent.

## 3. Results and Discussions 

### 3.1. Synthesis and Characterization of Semi IPNs CPSFTPU

#### 3.1.1. Characterization of Copolymers P(St-*co*-FMA)

Firstly, copolymerization of FMA and St was carried out with different monomer feeding ratios (St/FMA = 8/2, 6/4, 4/6, 2/8); the details are given in the experimental section. The structures of the obtained copolymers were confirmed by ^1^H NMR spectra as shown in Figure 3a. The signal around 5.05–4.49 ppm is attributed to the proton on -CH_2_-, and the peaks around 7.23–6.05 ppm originate from protons on benzene [38,39]. With the monomer feeding ratio of St/FMA decreasing from 8/2 to 4/6, the insertion ratio of St/FMA in copolymer chains is reduced from 37.5 to 6.13 as calculated from Table 1, giving more furan functional groups in the produced copolymer, which benefitted further reaction. When the monomer feeding ratio was decreased to 2/8, the insertion ratio of St/FMA in the copolymer achieved 1.31 providing the highest furan functional groups. However, the polymerization showed the lowest yield (45.3%) due to the insertion of polar FMA with furan group [40]. Therefore, the copolymer with sufficient furan groups obtained from a monomer feeding ratio of 4/6 was employed for network fabrication. According to ^1^H NMR spectra, P(St-*co*-FMA) copolymer was successfully obtained. In addition, the DSC analysis was performed on copolymers with different monomer feed ratios. As shown in Figure 3b, all curves of the copolymers showed a single glass transition temperature (*T*_g_) indicating the formation of copolymer rather than a mixture of tow types of homo-polymer. Moreover, the *T*_g_ of the copolymers increased from 44.7 to 78.9 ℃ with the ratio of FMA/St decreasing, suggesting less insertion of FMA.

#### 3.1.2. Characterization of Semi IPNs CPSFTPU

Semi IPNs were constructed by sequential pathway using TPU as the guest polymer, and CANs CPSF as matrix networks which form using boronic ester as crosslinker. FTIR spectra in Figure 4 confirmed this fact. In the FTIR spectrum of DHPM-B, the absorption peaks at 840 and 650 cm^−1^ can be ascribed to the stretch vibrations of =CH- in maleimide group [41]. Compared with the spectra of DHPM-B and CPSF/CPSFTPU samples, the absorption related to =CH- in CPSF/CPSFTPU completely disappeared. The absorption at 1391 cm^−1^ can be ascribed to the stretch vibrations of B−O [42,43]. Compared with the spectra of P(St-*co*-FMA) and CPSF/CPSFTPU, the absorption related to B−O in CPSF/CPSFTPU is improved, indicating the formation of networks. These observations suggest the occurrence of a chemical reaction between DHPM-B, P(St-*co*-FMA) and BDB. Moreover, the covalently cross-linked molecular architecture of the prepared CPSF/CPSFTPU samples can be further evidenced by the fact that they are insoluble in organic solvents such as toluene. The band observed at 1530 cm^−1^represents the N–H bending vibration of TPU [44]. Comparing CPSFTPU with TPU, the absorption at 1530 cm^−1^ sharply decreased, maybe due to hydrogen bond formation between C=O group in CANs and N–H groups in TPU chains [45]. Besides, the spectrum of CPSFTPU can be regarded as a simple overlap of the spectra of CPSF and TPU.

Typically, semi IPNs 6% CPSFTPU_1_ is composed of CANs matrix and linear TPU as the guest polymer. To study compatibility of CANs matrix and TPU in semi IPNs, morphology studies were performed. Figure 5 represents the SEM micrographs of the fractured surfaces of the semi IPNs 6%CPSFTPU_1_ sample with different magnification times. Almost homogeneous morphology of the sample is observed and TPUs are well dispersed in the CAN matrix in a micrometer scale with smooth interface and it is observed that there was no sea/island structure and phase interface in the 6%CPSFTPU_1_ polymer sample, which indicated that there is a good compatibility between CANs and TPU [46,47,48]. 

Equilibrium swelling experiments I showed that with the ratios of P(St-*co*-FMA)/TPU incensement, semi IPNs samples cannot be completely dissolved in THF, toluene, heptane or acetone and only achieved a certain degree of swelling in equilibrium swelling experiments. However, TPU or P(St-*co*-FMA) can completely dissolve in organic solvents. Similarly, with crosslinker content increasing, semi IPNs samples became hard to swell due to the cross-linked nature of networks. The cross-link density of semi IPNs is increased with improvement of the ratios of P(St-*co*-FMA)/TPU, and crosslinker content (0~6%) (as in Figure 6), revealing that a denser network is formed at higher ratios of P(St-*co*-FMA)/TPU and crosslinker content. Consequently, the sol fraction and swelling ratio are monotonously decreased with the ratios of P(St-*co*-FMA)/TPU, and crosslinker content (0~6%) increased. When crosslinker content changed to 9%, the sol fraction and swelling ratio were similar to those of 6% crosslinker content, because this achieved maximum cross-link density.

### 3.2. Dynamic Properties Analysis

The dynamic properties of the typical semi IPNs of 6%CPSFTPU_1_ were investigated by time- and temperature-dependent stress–relaxation analysis using DMA measurement as in Figure 7a. The characteristic relaxation time (τ*) that was determined at 1/e of the original stress [9,49], ranges from 607 s at 80 °C to 37.7 s at 140 °C, by the reason of the reversibility of DA reaction and boronic ester linker in the CAN matrix. As shown in Figure 7b, the temperature dependence of the relaxation time follows the Arrhenius law and it is demonstrated that the obtained semi IPNs present fluidity similar with inorganic glass. The activation energy of the 6%CPSFTPU_1_ was calculated to be 55.3 kJ mol^−1^ as reported previously [34,50,51]. The activation energy lies in between those of DA reaction-based CANs (104~111 kJ mol^−1^) [52,53] and those of borate exchange reaction-based vitrimers (around 26 kJ mol^−1^) [43,54]. Moreover, the introduction of linear TPU into CANs hinders both DA reaction and transesterification of boronic ester, leading to a higher activation energy of sim IPNs. Besides, according to the DMTA plots in Figure 7c, the 6%CPSFTPU_1_ sample shows a continually decreased storage modulus, as the temperature increased from 25 to 200 °C due to the presence of dissociated Diels−Alder adducts. Whereas, DMTA plots also displayed a rubbery plateau at 90~170 °C before network collapse, which is similar with associative vitrimers [55]. Consequently, semi IPNs should involve the reversible mechanism both associative and dissociated. The creep–recovery experiment corroborates that the maximum strain and the residual strain obviously increase with the tested temperature ranging from 80 to 140 °C due to the network rearrangement being easily activated at high temperature (Figure 7d) consistent with the stress–relaxation analysis. Therefore, the dynamic nature of DA chemistry and boronic ester in semi IPNs bestow the solid-state plasticity memories.

### 3.3. Mechanical and Thermal Properties

The representative tensile curves of semi IPNs 6%CPSFTPU_x_ with different ratios of P(St-*co*-FMA)/TPU and neat TPU are shown in Figure 8 and the data of mechanical properties is tabulated in Table 2. Neat TPU without networks is soft and relatively flexible, due to liner polymer chain structure. The Young modulus was measured to be 11.1 MPa, with a tensile strength of 9.7 MPa and strain at break of 651%. Strikingly, the mechanical properties of the semi IPNs 6%CPSFTPU_x_ series have been enhanced in comparison to TPU, because of the introduction of CANs with crosslinked structure. Moreover, with TPU addition increasing, Young modulus, tensile strength, and a strain at breaking of semi IPNs 6%CPSFTPU_x_ series firstly increased and then decreased. The semi IPNs 6%CPSFTPU_1_ sample shows the highest Young modulus measured to be 30.2 MPa, with a tensile strength of 28.4 MPa and a strain at breaking of 662% [56].

Figure 9 shows the tensile curves of semi IPNs with different crosslinker content and the data of mechanical properties are summarized in Table 3. Typically, with crosslinker content increasing from 0 to 6%, the mechanical properties of the semi IPNs x%CPSFTPU_1_ series continuously improve, Young modulus from 24.5 to 30.2 MPa, tensile strength from 7.5 to 28.4 MPa and strain at break from 550 to 662%. Because an increased crosslinker content brings out a more constrained CANs network and a restricted chain mobility due to increased cross-link density. Furthermore, crosslinker content increased from 6 to 9%; similar mechanical properties are obtained because of saturated crosslinking. The robust features of CANs and improvement of TPU impart improved mechanical properties of the semi IPNs CPSFTPU series.

The thermal stability of the TUP, BDB, DHPM-B, P(St-*co*-FMA), 0%CPSFTPU_1_ and 6%CPSFTPU_1_ are researched by TGA and the results are shown in Figure 10 and Table 4. Thermal stability factors, including initial decomposition temperature (the temperature of 5% weight loss, *T*_5*d*_) and the temperature of 30% weight loss (*T*_30*d*_) are determined by TGA. The values of *T_d_*_5_, *T*_30*d*_ and residual weight at 500/800 °C are summarized in Table 2. It was found that *T_d_*_5_ and *T_d_*_30_ values of 6%CPSFTPU_1_ are higher than these of 0%CPSFTPU_1_ and P(St-*co*-FMA). Moreover, residual weights at 500 and 800 °C of TPU and DHPM-B are rather small and even these of BDB both are 0.0%. However, residual weights at 500 and 800 °C of semi IPNs 6%CPSFTPU**_1_** are obviously higher than 0%CPSFTPU_1_ and P(St-*co*-FMA), because of the formation of network structures and the presence of TPU [45].

### 3.4. Self-Healing, Welding, and Shape Memory

Owing to the DA chemistry and transesterification reaction of boronic ester linkages in the CANs matrix, network rearrangement and bond reshuffling should take place, and covalent bonding can be re-established across the interfaces of the fractured surfaces. Consequently, it was proposed that semi IPNs can acquire self-healing through exchange-induced network rearrangement at high temperatures without adding healing agents. The schematically illustrates a self-healing mechanism of the sample based on a dynamic DA reaction and boronic ester bond as in Figure 11a. The thermally triggered self-healing mechanism of the 6% CPSFTPU_1_ sample has been investigated. A scratch recovery test was conducted using an optical microscope to monitor the sample before and after healing. In Figure 11c, the damaged 6% CPSFTPU_1_ film with 27.3-μm-wide scratches was firstly placed in a heating stage, and then the scratches on the damaged sample almost vanished after healing at 120 °C for 20 min in a convection oven, indicating excellent self-healing capability. To clarify the self-healing main from CANs rather than TPU, the same scratch treatment was performed on TPU. The width of scratches before and after healing are 28.2 and 16.3 μm respectively, it was obviously incompletely healed scratch in Figure 11d. Furthermore, the long strip of sample with width of 15 mm and thickness of 5 mm was cut in half with a razor blade, and then the two pieces were put together in 3 min at 120 °C under a pressure-free condition. Additionally, self-healing can be accelerated under pressure. As shown in Figure 11b, the welding film can lift a weight of 0.5 kg without breaking at the welded part, showing excellent welding properties because of the dynamic nature of CANs matrix [57].

It is inherently difficult to reshape permanently cross-linked semi IPNs; herein, the DA reaction as well as boronic ester undergoes associative transesterification in network of CANs matrix, which enables the semi IPNs network rearrangement and results in a gradual Arrhenius-like viscosity dependence, thus imparting the semi IPNs network an ability to be reshaped in a solid state. As a proof-of-concept, the strip-shaped sample of 6%CPSFTPU_1_ was first heated to 120 °C in a convection oven and turned into a U shape that could be fixed when the temperature was decreased to room temperature. When the temperature was increased to 120 °C again, the shape of “U” reverted to its original flat state very quickly within 2 min. The flat 6%CPSFTPU_1_ sample could further be reshaped into another shape (“O”/“8”/“S”), fixed, and then recovered by following the same “heating−cooling” procedure. Figure 12 shows the digital photos of the sample before and after shaping. The shape change is reversible and the process is repeatable. This is because the thermally induced dynamic DA reaction and boronic ester exchange reaction leads to the reorganization of CANs matrix cross-linked network in semi IPNs [58].

### 3.5. Reprocessing

In addition to the highly improved mechanical properties, shape-memory and self-healing capabilities, the semi IPNs also demonstrate another important property (reprocessability). The polymer networks of CANs matrix in semi IPNs are only exchanged without consumptions during recycling operation, therefore, several cycles might be achieved. To verify the reprocessability, 6%CPSFTPU_1_ sample was cut to small pieces and then reprocessed at 190 °C for 5 min by using a microinjection technique used for commercial thermoplastic; finally, homogeneous samples were obtained as in Figure 13a. The sample showed high thermal stability and remained unchanged, resulting from the nature of networks based on DA chemistry and boronic ester bond. Besides, recovery ratios of the mechanical properties for the recycled samples are shown in Figure 13c and Table 5. It is observed that most mechanical properties are restored after reprocessing. For example, the recovery ratios of tensile strength, Young’s modulus, and elongation at break for the sample are 95, 111 and 85%, respectively. A slightly decreased recovery rate of elongation at break and tensile strength of the 2nd generation is 70 and 82%, respectively. It is noted that there is an increased Young’s modulus in the 1st and 2nd generation samples and it might be originated from fully cured reprocessed sample due to the hot-pressing process compared with the original one [59]. Moreover, the 3rd generation sample can lead to the ultimate stretch of 430% as in Figure 13b. More strikingly, the samples can be repeatedly recycled because of robust characteristics of the DA reaction and dynamic boronic ester bonds in the CANs matrix. The manufactured sample in each cycle is shown in Figure 13a where the digit numbers indicate recycling generation. It is seen that small pieces of semi IPNs possess a good recycling ability in the same shapes. The stress–strain curves are shown in Figure 13a. It is remarkable to note that the sample still can achieve an ultimate stretch of 24 MPa even after three times of reprocessing. It also shows the evolution of the ultimate stretch as a function of the number of recycling. Although the properties of the recycled sample decay over recycling, it remains in a reasonably good range, confirming that DA reaction and boronic ester transesterification reaction play the critical roles in the reprocessing of the semi IPNs sample [60].

## 4. Conclusions

In brief, based on the duple dynamic DA adduct and boronic ester linker, a novel semi IPNs series have been prepared by a sequential reaction approach, using copolymer P(St-*co*-FMA), boronic ester crosslinker and TPU elastomer. The semi IPNs display a high compatibility between the CANs matrix and TPU gest polymer, providing excellent self-healing, recyclability, and shape memory properties. The thermally triggered self-healing of the semi IPNs has been achieved, giving a complete recovery of scratch sample using optical microscope as monitor. Through a simple microinjection technique, the cut samples can be reprocessed and mechanical properties of the recycled samples are well-restored. Taking advantage of the shape memory nature of the semi IPNs, a reversible and repeatable shape change is attained. Moreover, the DMA test is conducted to clarify the dynamic properties of semi IPNs. We anticipate that this novel strategy will pave way for the preparation of CANs with both excellent reconfigurability and functional applications.

## Figures and Tables

**Figure 1 polymers-13-01679-f001:**
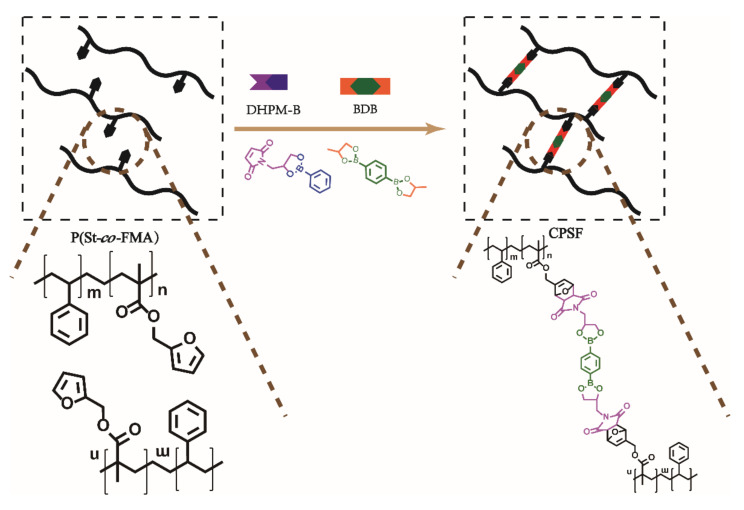
Preparation of CPSF CANs.

**Figure 2 polymers-13-01679-f002:**
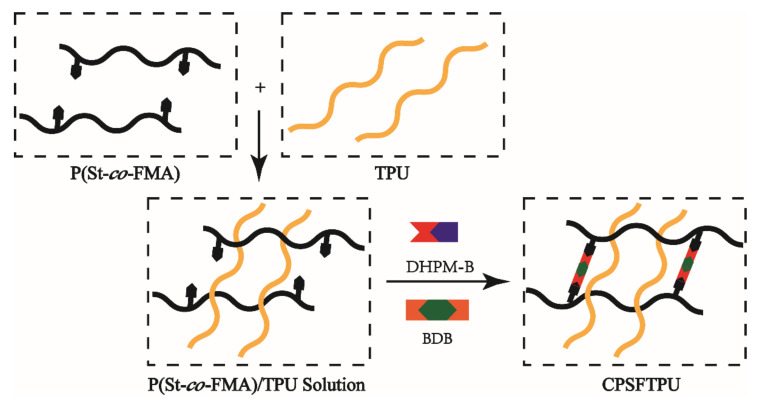
Preparation of semi IPNs of CPSFTPU.

**Figure 3 polymers-13-01679-f003:**
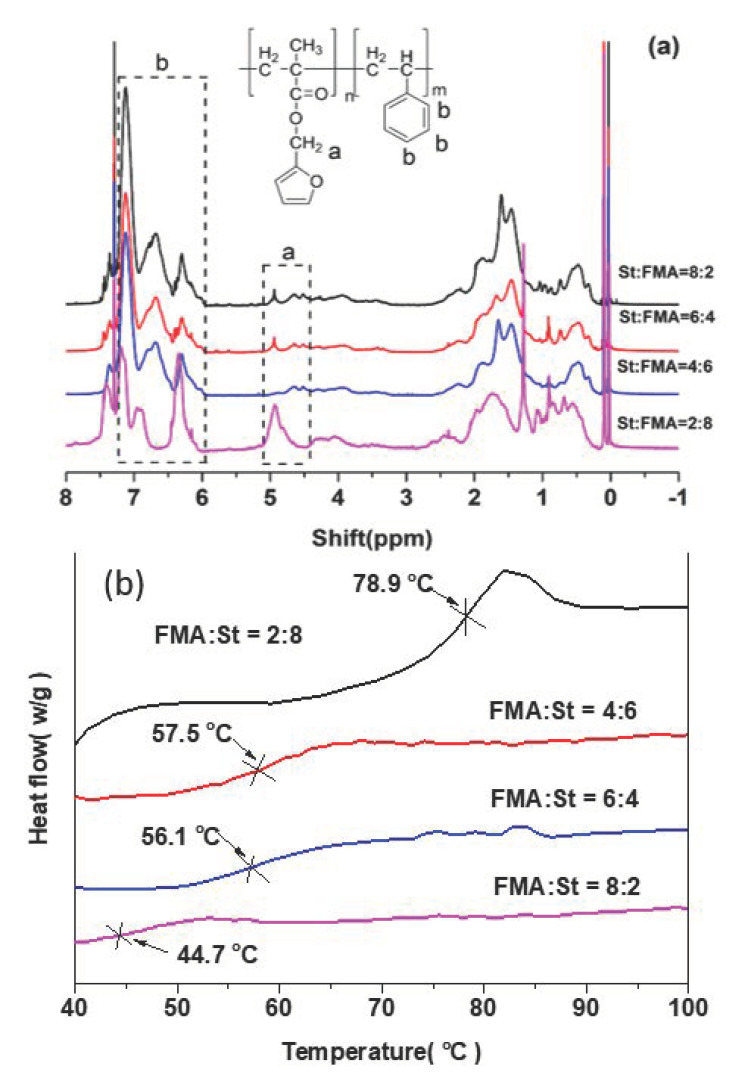
(**a**) ^1^H NMR spectra and (**b**) DSC curves of copolymer P(St-*co*-FMA) with different monomer feeding ratio.

**Figure 4 polymers-13-01679-f004:**
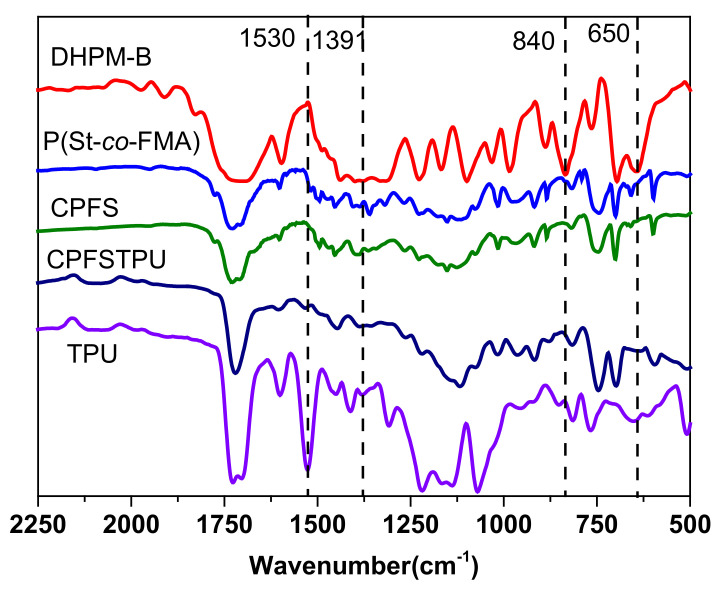
FTIR spectra of DHPM-B, P(St-*co*-FMA), CPSF, CPSFTPU and TPU.

**Figure 5 polymers-13-01679-f005:**
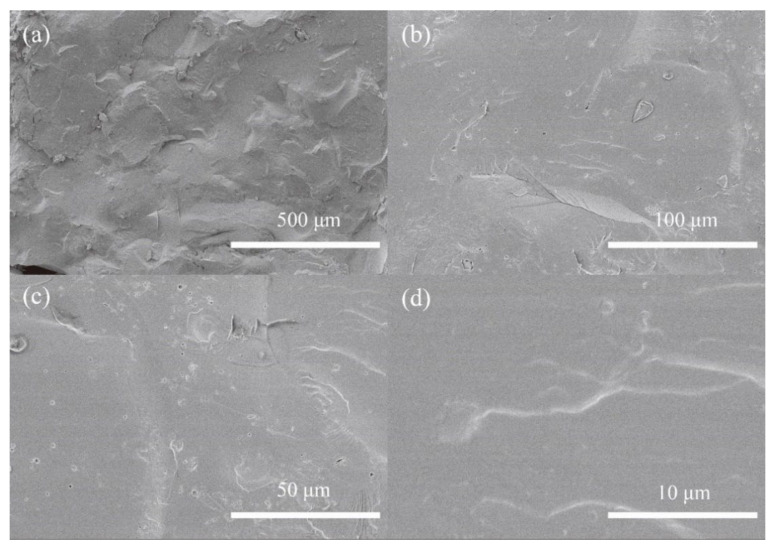
SEM images of semi IPNs 6%CPSFTPU_1_ with different magnification times (**a**) 500 μm; (**b**) 100 μm; (**c**) 50 μm; (**d**) 10 μm.

**Figure 6 polymers-13-01679-f006:**
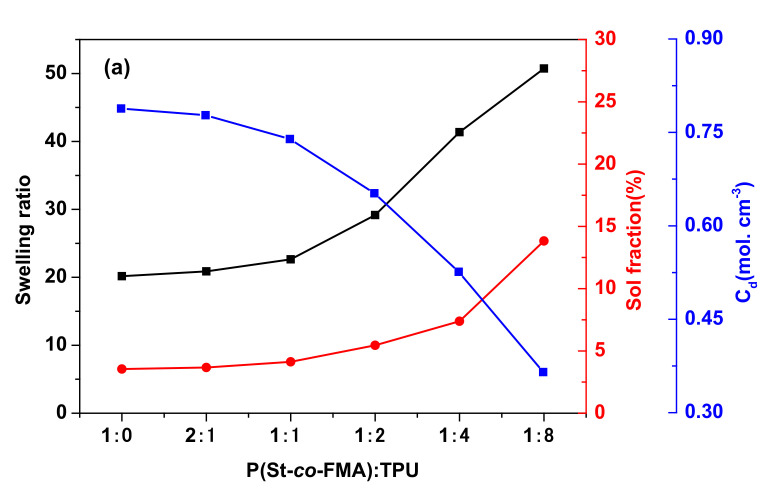
The swelling ratio, sol fraction and cross-link density of the semi IPNs series (**a**) with different ratios of P(St-*co*-FMA)/TPU, (**b**) with different crosslinker contents.

**Figure 7 polymers-13-01679-f007:**
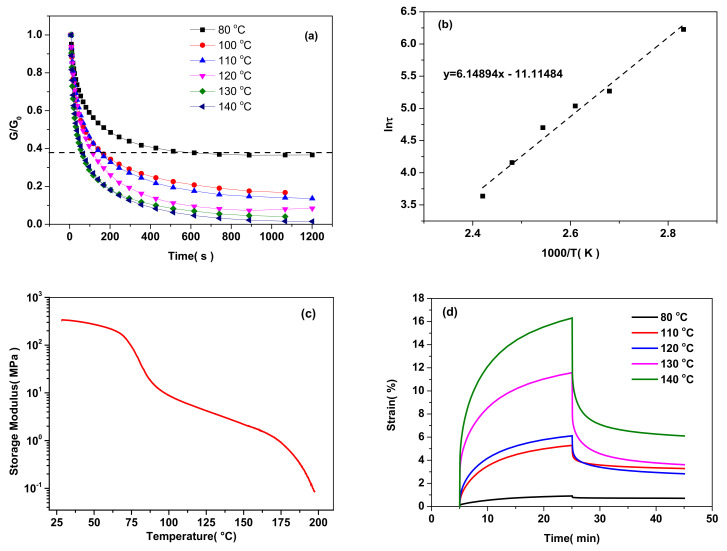
(**a**) Stress relaxation tests of the semi IPNs 6%CPSFTPU_1_ at different temperatures, (**b**) Arrhenius plot with linear fit. From this fit the activation energy is calculated to be 55.3 kJ mol^−1^, (**c**) DMTA trace with a continually decreased storage modulus of semi IPNs 6%CPSFTPU_1_, (**d**) creep tests at different temperatures.

**Figure 8 polymers-13-01679-f008:**
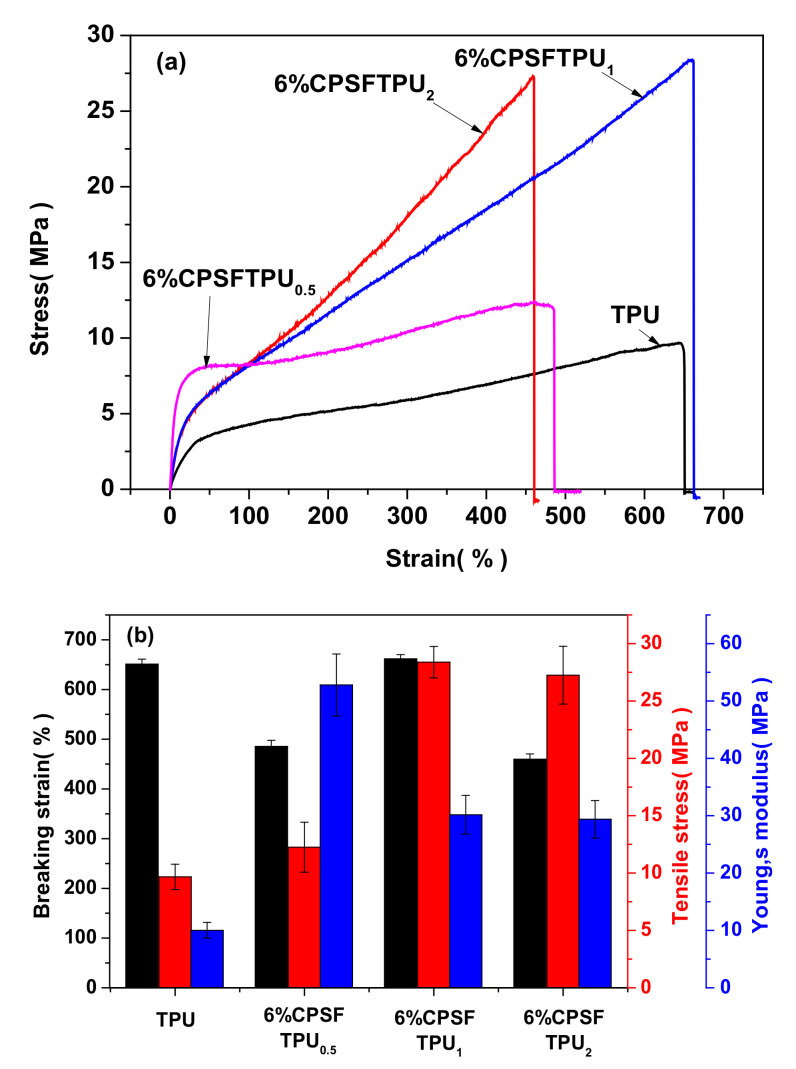
(**a**) Tensile curves and (**b**) mechanical properties of semi IPNs 6%CPSFTPU_x_ series with different ratio of P(St-*co*-FMA)/TPU.

**Figure 9 polymers-13-01679-f009:**
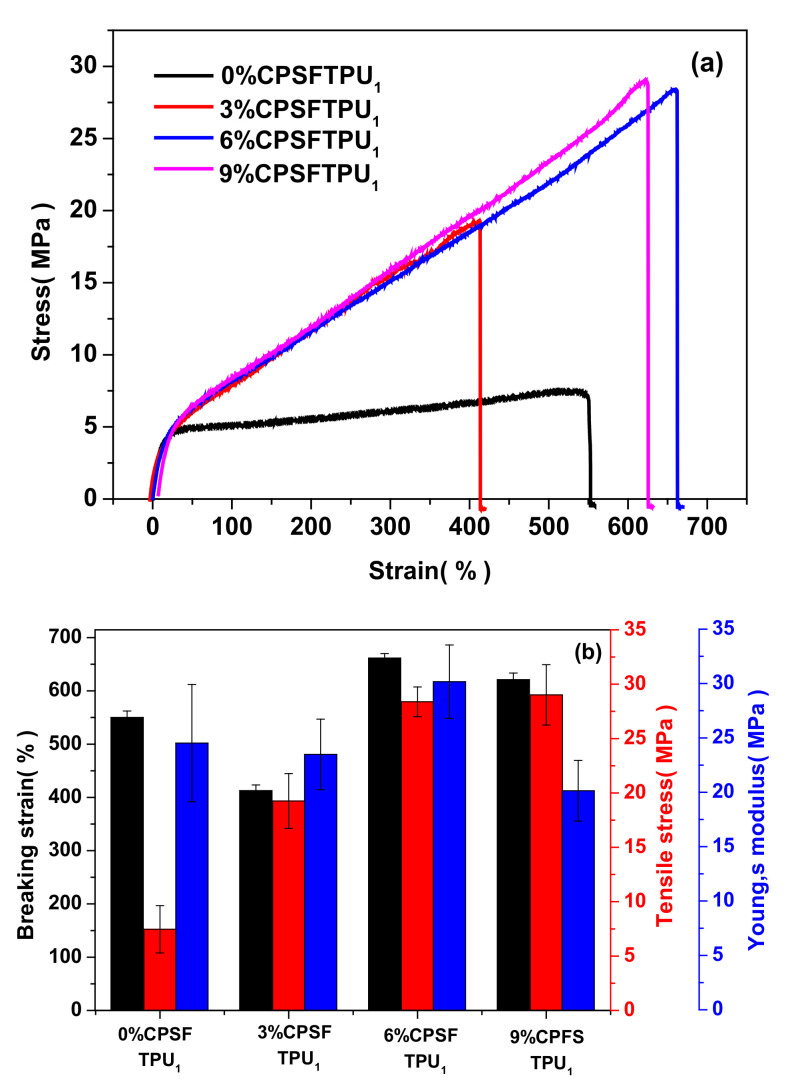
(**a**) Tensile curves and (**b**) mechanical properties of x%CPSFTPU_1_ series with different crosslinker content.

**Figure 10 polymers-13-01679-f010:**
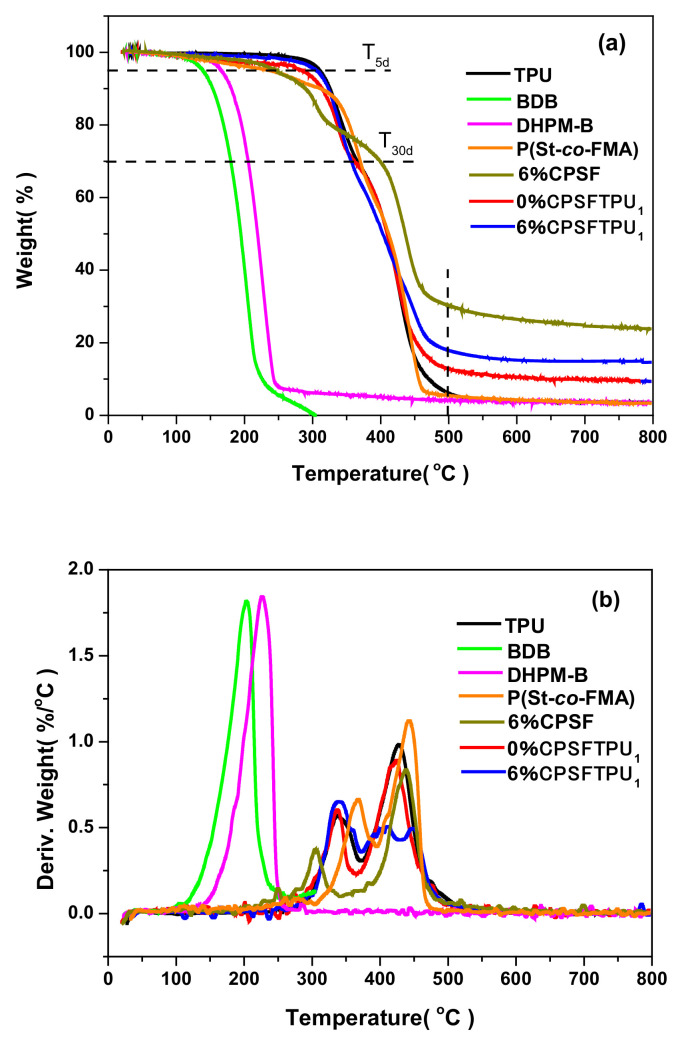
TGA (**a**) and DTG (**b**) curves of TPU, BDB, DHPM-B, P(St-*co*-FMA), 0%CPSFTPU_1_ and 6%CPSFTPU_1_.

**Figure 11 polymers-13-01679-f011:**
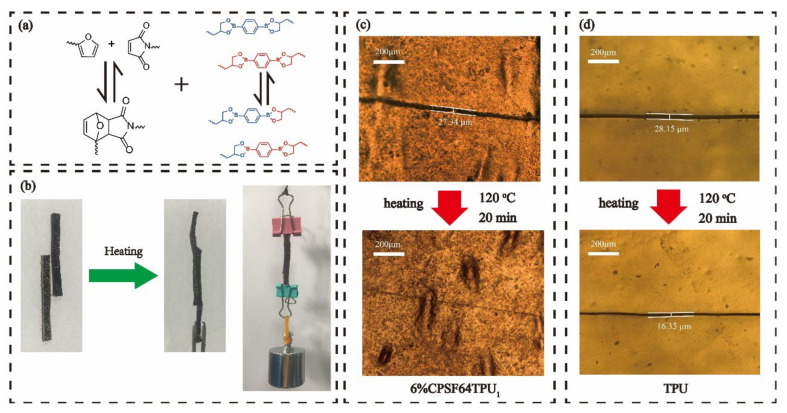
(**a**) Schematic representation of DA reaction (right) and the transesterification reaction of boronic ester linkages(right) (**b**) optical images of welding behaviors of 6%CPSFTPU_1_ sample, optical microscope images of the and TPU fracture surface before and after self-healing: (**c**) for 6%CPSFTPU_1_ and (**d**) for TPU.

**Figure 12 polymers-13-01679-f012:**
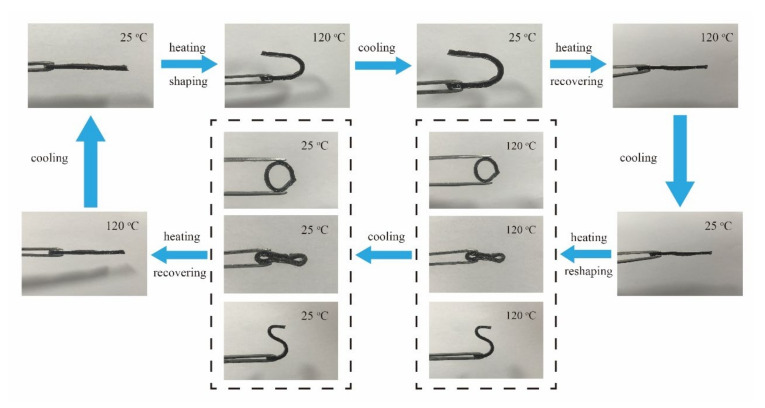
Optical images of shape-memory performance of the semi IPNs 6%CPSFTPU_1_.

**Figure 13 polymers-13-01679-f013:**
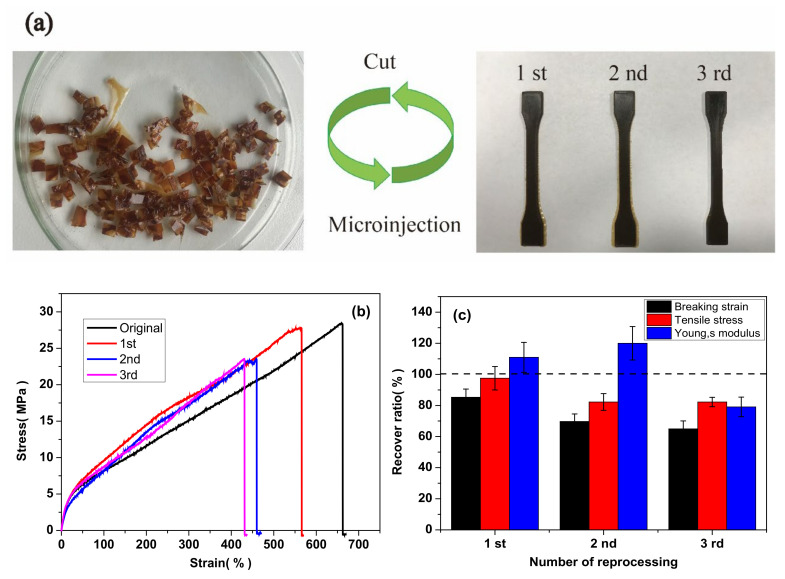
(**a**) Optical images of thermal recycling performance of semi IPNs 6%CPSFTPU_1_, (**b**) mechanical properties of the samples with different generations: tensile curves and (**c**) recovery ratios.

**Table 1 polymers-13-01679-t001:** Integration of proton on -CH_2_- or benzene groups in copolymers with different monomer feeding ratios.

Samples with Different Monomer Feeding Ratio (St/FMA)	Integration of Proton on –CH_2_–	Integration of Proton on Benzene
8:2	6.3	591.3
6:4	2.1	59.3
4:6	1.9	29.1
2:8	11.2	36.6

**Table 2 polymers-13-01679-t002:** Mechanical properties of semi IPNs 6%CPSFTPU_x_ series with different ratio of P(St-*co*-FMA)/TPU.

Sample	Breaking Strain (%)	Tensile Stress (MPa)	Young’s Modulus (MPa)
TPU	651.4 ± 10.11	9.7 ± 1.11	11.1 ± 1.35
6%CPSFTPU_0.5_	485.7 ± 12.18	12.3 ± 2.18	52.8 ± 5.39
6%CPSFTPU_1_	661.7 ± 8.37	28.4 ± 1.37	30.2 ± 3.38
6%CPSFTPU_2_	459.9 ± 10.52	27.3 ± 2.52	29.4 ± 3.25

**Table 3 polymers-13-01679-t003:** Mechanical properties of x%CPSFTPU_1_ series with different crosslinker content.

Sample	Breaking Strain (%)	Tensile Stress (MPa)	Young’s Modulus (MPa)
0%CPSFTPU_1_	550.1 ± 12.18	7.5 ± 2.18	24.5 ± 5.39
3%CPSFTPU_1_	412.8 ± 10.52	19.3 ± 2.52	23.5 ± 3.25
6%CPSFTPU_1_	661.7 ± 8.37	28.4 ± 1.37	30.2 ± 3.38
9%CPSFTPU_2_	621.2 ± 12.36	29.0 ± 2.77	20.1 ± 2.79

**Table 4 polymers-13-01679-t004:** Thermal stability factors of TUP, BDB, DHPM-B, P(St-*co*-FMA), 0%CPSFTPU_1_ and 6%CPSFTPU_1_ obtained from TGA and DTG curves.

Sample	*T*_5*d*_ (°C)	*T*_30*d*_ (°C)	Residual Weight at 500 °C (%)	Residual Weight at 800 °C (%)
TPU	295.3	350.5	5.5	3.6
BDB	141.3	179.7	0.0	0.0
DHPM-B	166.8	205.7	4.2	3.5
P(St-*co*-FMA)	225.7	369.7	5.8	3.3
0%CPSFTPU_1_	280.1	356.1	12.9	9.6
6%CPSFTPU_1_	305.5	356.8	17.7	14.5

**Table 5 polymers-13-01679-t005:** Recovery rate of mechanical properties of samples from different generations.

Generation	Breaking Strain Recovery Ratio (%)	Tensile Stress Recovery Ratio (%)	Young’s Modulus Recovery Ratio (%)
1st	85.4 ± 5.27	95.4 ± 7.52	110.9 ± 9.36
2nd	69.7 ± 4.81	82.2 ± 5.37	120.0 ± 10.74
3rd	65.0 ± 5.11	82.2 ± 3.02	79.1 ± 6.33

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
