# Peer review of "Dynamic Semi IPNs with Duple Dynamic Linkers: Self-Healing, Reprocessing, Welding, and Shape Memory Behaviors"

_polymers, 2021, doi:10.3390/polym13111679_

Round 1
Reviewer 1 Report
This paper investigates recycled polymers with covalent adaptable networks (CANs)by DA chemistry or the re-versible exchange boronic ester bonds. In details, the manuscript reports the novel type of CANs with multiple dynamic linkers based on a linear copolymer of styrene and furfuryl methacrylate and boronic ester crosslinker. A further variation of the microstructure was performed by taking into account the introduction of thermoplastic polyurethane to achieve a semi Interpenetrating Polymer Networks enhancing the properties of the CANs. The microinjection samples were mechanically tested and reshaped to assess the potentiality of reprocessing, welding and self-healing.
The INP based on IPNs based on CANs matrix and TPU as guest polymers is indeed a novel polymer structure worth looking at and investigate mainly for the potentiality of reshaping and self-healing performance.
The paper is accurate in the data description and analysis and the processing of examining the different P(St-co-FMA):TPU ratio and crosslinker content seem congruent with the overall final target of the investigation.
I would suggest to the author to revise the paper and add more analysis of the mechanical data related to the recovery of the material. In particular, the variation (positive for the 2nd processing) is not completely discussed. I would advise the author to formulate a rationale for the data presented in figure 9-10-1
3 taking into account the forming and reshaping microsctruture of the polymer in order to justify the found performance at 6% CPSFTPU1 which stands as the limiting concentration as well reported within the test at pag. 19.
The publication has indeed scientific and technical merit and it is worth publishing, I would suggest the author to look at the following issue for a minor revision.
Author Response
This paper investigates recycled polymers with covalent adaptable networks (CANs)by DA chemistry or the re-versible exchange boronic ester bonds. In details, the manuscript reports the novel type of CANs with multiple dynamic linkers based on a linear copolymer of styrene and furfuryl methacrylate and boronic ester crosslinker. A further variation of the microstructure was performed by taking into account the introduction of thermoplastic polyurethane to achieve a semi Interpenetrating Polymer Networks enhancing the properties of the CANs. The microinjection samples were mechanically tested and reshaped to assess the potentiality of reprocessing, welding and self-healing.
The INP based on IPNs based on CANs matrix and TPU as guest polymers is indeed a novel polymer structure worth looking at and investigate mainly for the potentiality of reshaping and self-healing performance.
Comment 1: The paper is accurate in the data description and analysis and the processing of examining the different P(St-co-FMA):TPU ratio and crosslinker content seem congruent with the overall final target of the investigation.
Authors response: We thank the referee for his/her comments.
Comment 2: I would suggest to the author to revise the paper and add more analysis of the mechanical data related to the recovery of the material. In particular, the variation (positive for the 2nd processing) is not completely discussed. I would advise the author to formulate a rationale for the data presented in figure 9-10-1
Authors response: We have added the more analysis of the mechanical data in Table 2, 3 and 5, and have added more discussion about 2nd processing in the revised manuscript.
Comment 3: taking into account the forming and reshaping microsctruture of the polymer in order to justify the found performance at 6% CPSFTPU1 which stands as the limiting concentration as well reported within the test at pag. 19.
Authors response: Yes, we selected the 6% CPSFTPU1 with the best properties for self-healing, shape memory and reprocessing. The network can rearrange by dynamic AD adducts and bionic ester bond, thus the network almost unchanged after self-healing, shape memory and reprocessing, which confirmed by the phenomenon of without changed sample color after self-healing, shape memory and reprocessing.
Reviewer 2 Report
This work continues the line of research first proposed by Leibler on the example of vitrimers. It is devoted to the experimental study of polymers with covalently adaptable networks having dynamic linkers. The synthesis and dynamic properties of the obtained semi Interpenetrating Polymer Networks are described. The developed technology makes it possible to obtain polymers as a result of reprocessing and their self-healing and welding, as well as with shape memory. These results are of undoubted interest and can be recommended for publication in the journal.
Author Response
We thank the referee for giving his/her comments.